# Effects of Oleuropein and Hydroxytyrosol on Inflammatory Mediators: Consequences on Inflammaging

**DOI:** 10.3390/ijms24010380

**Published:** 2022-12-26

**Authors:** Fanny Pojero, Anna Aiello, Francesco Gervasi, Calogero Caruso, Mattia Emanuela Ligotti, Anna Calabrò, Antonio Procopio, Giuseppina Candore, Giulia Accardi, Mario Allegra

**Affiliations:** 1Laboratory of Immunopathology and Immunosenescence, Department of Biomedicine, Neurosciences and Advanced Technologies, University of Palermo, 90134 Palermo, Italy; 2Specialistic Oncology Laboratory Unit, ARNAS Hospitals Civico Di Cristina e Benfratelli, 90127 Palermo, Italy; 3Department of Health Sciences, University Magna Graecia of Catanzaro, 88100 Catanzaro, Italy; 4Department of Biological, Chemical and Pharmaceutical Sciences and Technologies, University of Palermo, 90123 Palermo, Italy

**Keywords:** oleuropein, hydroxytyrosol, inflammaging, nutraceutical

## Abstract

Aging is associated with a low-grade, systemic inflammatory state defined as “inflammaging”, ruled by the loss of proper regulation of the immune system leading to the accumulation of pro-inflammatory mediators. Such a condition is closely connected to an increased risk of developing chronic diseases. A number of studies demonstrate that olive oil phenolic compound oleuropein and its derivative hydroxytyrosol contribute to modulating tissue inflammation and oxidative stress, thus becoming attractive potential candidates to be used in the context of nutraceutical interventions, in order to ameliorate systemic inflammation in aging subjects. In this review, we aim to summarize the available data about the anti-inflammatory properties of oleuropein and hydroxytyrosol, discussing them in the light of molecular pathways involved in the synthesis and release of inflammatory mediators in inflammaging.

## 1. Introduction

The rate of non-communicable chronic diseases (including cardiovascular, metabolic, and neurodegenerative diseases as well as cancer) tends to increase with age [1,2,3], in parallel with a decline in the performance of immune defenses [4]. In fact, aging leads to the accumulation of both exhausted and senescent immune cells, the latter process being qualified as “immunosenescence” [2,4,5,6]. During aging, leukocytes exhibit multiple defects in their ability of mounting effective immune responses against external and internal aggressors, related to (I) altered recirculation profiles and functional impairment of both innate and adaptive immune cells, and (II) the contraction of antigen receptor diversity repertoires and expansion of memory pools under the pressure of chronic antigenic load specifically in the adaptive immune branch [2,5,6,7,8,9,10,11,12]. Immunosenescence is strictly interconnected with a state called “inflammaging”, but more recent progresses suggest that the two phenomena deserve a separate dissertation [4,9,12].

The term “inflammaging” was proposed to describe an aging-related low-grade, systemic chronic inflammation detectable as the rise of multiple serum inflammatory biomarkers, such as interleukin 1 (IL-1), IL-6, IL-8, IL-18, and tumor necrosis factor-α (TNF-α) in the absence of an appropriate secretory stimulus, paving the way for altered tissue homeostasis and increased oxidative stress, both being key players in chronic diseases [3,5,6,9,12,13,14,15,16,17].

Senescence-associated secretory phenotype (SASP) represents the secretome of senescent cells and includes the pro-inflammatory cytokines that are increased in inflammaging together with degradative enzymes such as matrix metalloproteinases (MMPs) [3,11,15,18]. However, together with senescent cells, immune cells (especially throughout the immunosenescence processes) also account for the production of inflammatory mediators during inflammaging [2,4,9,12,13], as summarized in Figure 1. The production of inflammatory mediators, e.g., IL-1, IL-6, and IL-8, further promotes cellular senescence and consequently hampers the resolution of inflammation [9,19] (Figure 1). In addition, aged immune cells are not efficient in clearing senescent cells, thus alimenting a vicious cycle [9].

A number of transcription factors appear to drive both SASP and inflammaging, but the best characterized role is surely that of nuclear factor kappa-light-chain enhancer of activated B cells (NF-κB). This transcription factor is kept inactive in the cytosol bound to NF-κB inhibitor alpha (IκBα) and becomes activated after an inflammatory stimulus-mediated phosphorylation, ubiquitination and degradation of IκBα, and NF-κB phosphorylation and nuclear translocation. NF-κB nuclear localization and activity are increased during aging and directly account for the expression of inflammatory mediators [20,21,22,23,24,25]. Some emerging evidence confirms the role of other signaling patterns in inflammaging, as described for the mitogen-activated protein kinase (MAPK) pathway [26,27,28]. Some of the involved kinases, such as mitogen-activated protein kinase (MEK), are deputed to the propagation of stress and inflammatory stimuli, while others, such as extracellular signal-regulated kinase (ERK, also known as p44/42 MAPK), c-Jun N-terminal kinase (JNK), and p38 MAPK, account for diverse inflammation-promoting effects, including direct phosphorylation of transcription factors ruling the expression of pro-inflammatory mediators [29,30].

The exact mechanisms underlying inflammaging have not been fully characterized yet, but both concomitant genetic and environmental/lifestyle-related factors (encompassing insufficient physical exercise, drug and food abuse, and smoking) appear to be involved, allocating a very important role to reactive oxygen species (ROS) [2,3,9,12,13,14,31,32]. During aging, redox homeostasis is compromised due to the mitochondrial impairment leading to ROS accumulation that overcomes cellular abilities to cope with oxidative stress and drives the overexpression of pro-inflammatory genes in a vicious cycle that reinforces oxidants production [3,14,31,33,34]. In addition, systemic inflammation and oxidative stress cooperate to cause a block in the cell cycle. When this phenomenon is coupled with the activation of the mammalian target of rapamycin (mTOR) pathway, it leads to a permanent loss of the ability to restore cell proliferation, hypertrophy, and the development of the hallmarks of cellular senescence, within a process called geroconversion [34,35,36]. Finally, mTOR contributes to the activation of NF-κB, with a further exacerbation of the inflammatory state [20,24].

Another potential contributor to inflammaging may be represented by gut-microbiome-derived lipopolysaccharide (LPS). Age-associated gut dysbiosis and alteration of the intestinal barrier configure a degenerative frame characterized by the increase in circulating LPS [37,38,39]. Gut-derived LPS strengthens the production of inflammatory cytokines and contributes to the insurgence and persistence of inflammaging [12,32,33,34,40].

It becomes evident that immunomodulating and/or antioxidant molecules become fundamental to prevent an unbearable oxidative stress, limit inflammaging, and promote healthy aging [14,31].

Diet and dietary supplements may be beneficial in modulating the secretion of inflammatory cytokines [9,41,42]. The anti-inflammatory effect of the Mediterranean diet has largely been documented [9,41,42], and a crucial role for dietary olive oil phytochemicals as immune-modulators and antioxidants has recently emerged [42,43,44,45].

Among the phenolic compounds characterized in olive oil, secoiridoid oleuropein (OLE) and its derivative hydroxytyrosol (3,4-dihydroxyphenylethanol, HT) [43,45,46,47,48,49,50] have largely been studied for their multiple protective effects against cardiovascular diseases, cancer, and infections (as detailed in [44,46,47,48,49,50]). OLE represents one of the two major secoiridoids synthetized by olive (*Olea europaea* L.) and can be found in both leaves and drupes [43,48,49]; it is the ester formed by tyrosol (p-hydroxy-phenylethyl alcohol, HPEA) and the glycosidic derivative of elenolic acid, also existing in its aglycone form [43,47,48,49,50,51]. HT is obtained from the hydrolysis of OLE during olive maturation and olive oil storage [43,47,49,50,51]. In both in vitro and in vivo scenarios, OLE and HT interfered with the production of inflammatory mediators [46,49]. However, little is known about the anti-inflammatory action throughout inflammaging. In this review, we summarize the effects of OLE and HT on the synthesis and action of both inflammatory mediators overexpressed in inflammaging and anti-inflammatory cytokines. Experimental evidence is analyzed in the light of cellular pathways ruling low-grade chronic inflammation that are specifically deregulated in aged subjects. The possible uses of OLE and HT to disrupt molecular patterns leading to altered ROS homeostasis and sterile inflammation during aging will also be discussed.

## 2. OLE and HT Effects on Inflammatory Cytokine and on Redox Homeostasis

Available studies provide some essential pieces of information about the action of OLE and HT on pro- and anti-inflammatory circuits at both local and systemic level. Although both phenols share some similarities in their mechanism of action, some important differences may be detected about the net elicited effects according to the considered cytokine and the experimental scenario, as commented in the following paragraphs. The lists of in vitro and in vivo effects of OLE and HT on the synthesis and release of inflammatory mediators (as reported in this review) are indicated in Table 1 and Table 2, respectively.

### 2.1. IL-1

Actually, the acronym IL-1 is used to indicate two cytokines encoded by separated genes and secreted by both innate and adaptive immune cells: IL-1α and IL-1β [16,102,103,104,105]. Both of them are able to solicit similar inflammatory responses through binding to the same receptor and trigger NF-κB signaling [103,104,105]. IL-1α works as a molecule of the damage-associated molecular patterns (DAMP), being released by necrotic cells, thus promoting immune cell infiltration [16,104,105]. IL-1β is secreted during host aggression by pathogens in response to the assembly of the inflammasome [16,104,105]. Both IL-1α and IL-1β are able to feed their own production and release [16,104,105].

In older subjects, increased IL-1 levels are associated with an augmented risk of mortality and morbidity, including cancer, atherosclerosis, type 2 diabetes, and osteoporosis [16,102,104,105].

As demonstrated in vitro, OLE acts on both IL-1β release and IL-1β-mediated inflammatory action. OLE-rich extract (OLE concentration = 379 mg/g) reduced IL-1β expression in LPS-stimulated RAW264.7 cells in a time-dependent manner [52]. This piece of data differs from a report documenting that OLE was not effective in reducing IL-1β expression in LPS-challenged murine RAW264.7 cells. However, this may represent the result of a shorter exposure to OLE (6 or 18 h vs. 24 or 72 h in the first mentioned study) [53]. OLE in its glycoside form decreased IL-1β production at low concentrations (0.1 μM) in whole blood cell cultures in the presence of LPS [54]. Similarly, 10 nM and 1 μM OLE reduced IL-1β release by osteoarthritic chondrocytes [55]. As documented in LPS-exposed RAW264.7 macrophages, pre-treatment with 300 μM OLE produced a reduction in IL-1β at mRNA and protein levels by inhibition of both phosphorylation of IκB-α and nuclear translocation of NF-κB [56]. Similarly, OLE counteracted IL-1β-induced inflammation via suppression of NF-κB and MAPK signaling [106].

In vitro data for HT suggest that timing is a fundamental factor in determining the final outcome of HT administration when the inflammatory stimulus is LPS. Experimental results obtained in vitro from LPS-challenged RAW264.7 macrophages demonstrated that 25 μg/mL and 12.5 μg/mL HT cotreatment for 24 h increased IL-1β release, with a mechanism that may depend on IκB-α degradation and NF-κB activation [57]. Instead, 5 μM, 10 μM, 25 μM, 50 μM, and 100 μM HT cotreatment for 6 or 18 h produced no change in IL-1β expression in the same cell line undergoing an analogous stimulation [53]. By contrast, pre-treatment with HT (50 or 100 μM) significantly reduced IL-1β at both mRNA and protein level after LPS stimulation of RAW264.7 cells with a mechanism involving a reduction in ERK phosphorylation [58]. When 41 μM HT was used on human monocytes after LPS stimulation, it downregulated IL-1β expression and reduced the cytokine secretion [59]. The timing of HT use (before, during, or after the incubation with inflammatory stimuli) was not relevant when the pro-inflammatory trigger was something different from LPS. In rat chondrocytes stimulated with TNF-α in the presence of HT, 25 μM to 100 μM HT reduced TNF-α-induced IL-1β release [60]. In human PBMCs, as low as 30 min of HT pre-treatment was sufficient to prevent the increase in IL-1β induced by oxysterol exposure [61].

In vivo, OLE reduced serum IL-1β in LPS-induced sepsis, decreasing NF-κB mRNA levels [80]. Similarly, in a myocardial ischemia/reperfusion model, the reduction in serum IL-1β implied a decrease at protein level in phospho-IκBα (p-IκBα, an inhibitor of NF-κB), kinases phospho-MEK (p-MEK) and phospho-ERK (p-ERK), and cell survival/apoptosis ruling transcription factor p53 [81]. An analogous reduction in IL-1β levels elicited by OLE was obtained in serum in a heart failure model [82], and at tissue level in a model of airway inflammation triggered by cigarette smoke [83] and in dextran sodium sulphate (DSS)-induced chronic colitis [84]. Similarly, in a model of acetic-acid-induced ulcerative colitis, OLE downregulated IL-1β expression in colon tissue [85]. Experiments performed in a model of high fat diet showed that olive leaf extracts (containing 10% OLE)-mediated IL-1β reduction depends on a regulation at transcriptional level at least in liver and adipose tissues [86].

In an in vivo model of pristane-induced systemic lupus erythematosus, 100 mg/kg HT was able to reduce IL-1β secreted by LPS-stimulated splenocytes and macrophages [87]. Further studies on renal tissue in the same model defined a HT-mediated mechanism that may prevent the degradation of IκBα, phosphorylation of MAPK, and nuclear translocation of p65 (a subunit of NF-κB) [87]. A report on apoE-/- mice confirmed that a HT-mediated reduction in the phosphorylated forms of p38 MAPK and NF-κB in liver was recorded together with a reduction in serum IL-1β levels [88]. Similarly, in an in vivo model of acute liver injury, HT downmodulated IL-1β in liver tissue [58].

### 2.2. IL-6

IL-6 is a pleiotropic interleukin, which acts as a pro-inflammatory factor when produced by senescent and immune cells on NF-κB- and TNF-α-dependent pathways [9,19]. IL-6 levels are associated with age, morbidity, mortality, and high serum C reactive protein (CRP) levels [3,9,102]. However, IL-6 acts as a myokine (anti-inflammatory hormone-like mediator) when produced by muscles after physical exercise [9].

In vitro, the effects of OLE on IL-6 synthesis and release were model- and dose-dependent, whereas results for HT were reproduced across the different models tested. Long-term treatment (4–6 weeks) up to senescence of pre-senescent human fetal lung fibroblast and human neonatal lung fibroblast with 1 μM HT and 10 μM OLE aglycone determined a reduction in IL-6 release vs. senescent untreated cells, with statistically significant effects exhibited by HT in both cellular models [62]. Instead, low-dose (0.1 μM) OLE glycoside had no effect on IL-6 production when used to treat human whole blood cell cultures sampled from young male donors and stimulated with LPS [54]. By contrast, low concentrations of OLE (100 nM and 1 μM) reduced IL-6 production by osteoarthritic chondrocytes [55]. Similarly, in LPS-stimulated RAW264.7 macrophages, OLE-rich extract and 5 μM, 10 μM, and 20 μM OLE reduced IL-6 mRNA and cytokine release, respectively [52,63]. 30 μM and 100 μM OLE also decreased IL-1β-induced release of IL-6 by adult retinal pigment epithelium [64].

As anticipated, in vitro pre-treatment with 50 μM and 100 μM HT efficiently downregulated IL-6 expression in and cytokine release by LPS-stimulated RAW264.7 macrophages [58]. In human monocytes stimulated with LPS, 41 μM HT diminished IL-6 at both mRNA and protein level [59]. In rat chondrocytes, 25 μM, 50 μM and 100 μM HT reduced TNF-α-induced IL-6 secretion [60].

Molecular mechanisms elicited by OLE and HT seem to be strictly related to the pro-inflammatory stimuli triggering IL-6 secretion. Both HT and OLE reduced TNF-α stimulated release of IL-6 in murine osteoblast-like cells with a mechanism that might be related to the suppression of TNF-α-induced phosphorylation of p44/p42 MAPK and AKT (a serine/threonine kinase involved in ROS homeostasis), with HT being also able to mediate the downregulation of IL-6 transcription [65,107,108]. Instead, in LPS-stimulated macrophages and in γ-irradiated neonatal human dermal fibroblasts, OLE and HT reduced the expression of IL-6 at both mRNA and protein levels by diminishing p38 MAPK phosphorylation, and NF-κB phosphorylation and translocation [56,66,67]. Additionally, OLE interferes with LPS promoted toll-like receptor 4 (TLR4) dimerization, thus alleviating inflammation through the impairment of the TLR4-MyD88-NF-κB/MAPK axis [66].

Data recorded in vivo for an OLE-mediated effect on circulating IL-6 led to inconclusive results. In an in vivo model of acute pancreatitis, OLE showed no prophylactic effect in terms of reduction in serum IL-6 [89], but in LPS-induced sepsis [80], in cyclophosphamide and epirubicin combined treatment toxicity [90], in myocardial ischemia/reperfusion [81], and in experimental autoimmune myocarditis [91], an OLE anti-inflammatory effect manifested in part as a decrease in IL-6 serum level. An analogous reduction in plasma IL-6 levels was mediated by OLE in a model of sepsis-induced myocardial injury [92]. In the limits of biological differences of used animals and experimental models, discrepant results cannot be explained in terms of the variability of the administered doses, since in acute pancreatitis, experimental autoimmune myocarditis, and sepsis-induced myocardial injury models, researchers employed a 20 mg/kg dose of OLE, with 25 and 50 mg/kg doses used only in LPS-induced sepsis [80,89,91,92].

As regards the effect of HT on serum IL-6 levels, in vivo data demonstrated that 200 mg/L HT reduced circulating IL-6 levels in cyclophosphamide-induced immunosuppressed broilers [94], and a reduction in serum IL-6 was determined by 10 mg/Kg HT in apoE-/- mice [88].

Data recorded in vivo at tissue level demonstrated that results were consistent across models and used doses for both OLE and HT. 1 mg/Kg, 10 mg/Kg, and 25 mg/kg olive leaf extract containing 10% OLE downmodulates IL-6 transcription in liver and adipose tissue [86]. 50 mg/Kg and 100 mg/kg OLE reduced IL-6 concentration in a model of azoxymethane (AOM)/DSS-induced colorectal cancer (CRC) [93]. In a model of pristane-induced systemic lupus erythematosus, administration of 100 mg/kg HT reduced the amount of IL-6 secreted by LPS-stimulated splenocytes and macrophages [87]. Further, in an in vivo model of acute liver injury, HT reduced IL-6 expression in liver tissue [58].

It appears that in vivo and in vitro OLE- and HT-mediated reduction in IL-6 is flanked by the blockade of NF-κB, ERK, JNK, and p38 MAPK pathways at local level [64,80,87,91].

From a pharmacokinetic point of view, some experimental proofs seem to suggest that OLE may not need an intracellular vehicle and/or ligand to exert its anti-inflammatory function on IL-6 production. A study by Huguet-Casquero et al. compared the performance of OLE as a suspension in deionized water (OLEsus) and in the form of nanostructured lipid carrier loaded with OLE (NLC-OLE) in an in vivo model of acute colitis. The authors concluded that the performance of OLEsus was superior to that of NLC-OLE in reducing IL-6 levels vs. controls [68].

Experimental use of OLE on human overweight patients led to an increase in serum IL-6, leaving some concerns about the safety of OLE supplementations in humans [109].

### 2.3. TNF-α

TNF-α is a transmembrane protein expressed by numerous different cell types (including macrophages, dendritic, and senescent cells, among others) that is cleaved by tumor necrosis factor converting enzyme (TACE) and released as a consequence of inflammatory stimuli. Both (transmembrane and soluble) forms bind with different affinity to TNF-α receptors TNFR1 (CD120a) and TNFR2 (CD120b) and promote a pro-inflammatory environment by activating NF-κB and MAPK and ruling the synthesis of both adhesion molecules and soluble mediators. TNF-α also accounts for the regulation of proliferation and apoptosis/necrosis [11,110,111,112]. In fact, after binding to TNFR1 and the recruitment of TNFR1-associated DD (TRADD), at the TNFR1 death domain (DD), TNF-α may promote cell survival via NF-κB and MAPK activation or cell death by apoptosis (via caspase 8) and necroptosis (through mixed lineage kinase domain-like protein (MLKL)). In such a case, cell fate strictly depends on microenvironmental conditions, and ruling mechanisms are not fully understood. Instead, by binding to TNFR2 and the recruitment of TNFR-associated factor (TRAF) 1 and 2, TNF-α promotes cell survival and inflammation through the mechanisms described above [110,111,112]. TNF-α production tends to increase with age, with the highest circulating levels found in successful aging, despite elevated plasma levels being described as associated with mortality and age-related diseases (such as cardiovascular diseases, obesity, and insulin resistance) by some authors [11,102,110,111,112,113,114,115].

The in vitro action of OLE on TNF-α secretion may strictly be model-dependent. Murine J774 macrophages were pre-treated with OLEsus, NLC-OLE, and blank (unloaded) nanoparticles (NLC-BLANK) before stimulation with LPS. While OLEsus failed in reducing TNF-α, nanoparticles decreased TNF-α levels independently from the carrying of OLE, thus excluding a biological meaning of OLE for such a reduction in this experimental context [68]. By contrast, 5 μM, 10 μM, and 20 μM OLE reduced TNF-α secretion in LPS-stimulated RAW264.7 macrophages [63], but OLE-mediated effects on TNF-α transcription were not evident if treatment lasted less than 24 h and OLE was used as a cotreatment together with LPS [53]. Treatment with low-dose OLE glycoside (0.1 μM) did not alter TNF-α production when human whole blood cell cultures were stimulated with LPS [54], while the same dose of OLE was sufficient to cause a decrease in TNF-α release in osteoarthritic chondrocytes [55]. Consistently, in human PBMCs, OLE exerted a dose-dependent suppression of LPS-elicited TNF-α secretion [69].

Instead, in vitro data for HT revealed the activation of a pro-inflammatory molecular network when HT was not used as a pre-treatment before exposing cells to LPS. Cotreatment of LPS-stimulated mouse RAW264.7 macrophages with 50 μg/mL and 25 μg/mL caused an increase in TNF-α production flanked by a HT-dependent NF-κB activation [57], but an exposure to HT shorter than 24 h failed in producing an upregulation of TNF-α [53]. A species-dependent effect may be suspected, since in mouse spleen lymphocytes, treatment with 12.5 μg/mL and 6.25 μg/mL HT was per se able to increase TNF-α production [57]; however, analogous evidence was reported for human cells, too. Coincubation of LPS-stimulated human monocytes with 50 μM and 100 μM HT upregulated TNF-α and increased the cytokine secretion through a mechanism mediated by a reduction in intracellular cAMP [70,71]. By contrast, use of HT as a pre-treatment led to completely different results in both murine and human cells. Pre-treatment with 50 μM and 100 μM HT was able to reduce expression and release of TNF-α in LPS-challenged RAW264.7 cells [58]. Treatment of human monocytic cell line THP-1 with HT for 10 min before LPS stimulation led to a reduction in TNF-α transcription and cytokine production with a dose-dependent pattern [72]. Similarly, pre-treatment of human colorectal cancer cell lines with HT before exposing cells to LPS caused a marked reduction in both mRNA and protein TNF-α levels by reducing phospho-NF-κB [73]. When 41 μM HT was used after LPS stimulation of human monocytes, it led to results analogous to those obtained when employed as a pre-treatment: it reduced TNF-α mRNA and cytokine secretion [59].

Data obtained in vivo deserve a special dissertation regarding the possible local rather than systemic action of OLE. Results for an in vivo model of acute colitis showed that both OLEsus and NLC-OLE significantly reduced TNF-α release in colonic tissue, with the lowest values detected for OLEsus [68]. Further, in in vivo models of AOM/DSS-induced CRC [93], unilateral ureteral obstruction [95], and cisplatin-induced acute renal injury [96], OLE reduced tissue TNF-α [93,95,96]. Similarly, an in vivo model of high-fat diet revealed that 1 mg/Kg, 10 mg/Kg, and 25 mg/kg olive leaf extract (containing 10% OLE) downmodulate TNF-α transcription in liver and adipose tissue [86], and that orally administrated OLE-rich extracts are able to reduce TNF-α levels in plasma and liver [97]. Further confirmations of OLE effects on circulating cytokine levels raised from studies on LPS-induced sepsis [80], cyclophosphamide, and epirubicin combined treatment toxicity [90], heart failure [82], myocardial ischemia/reperfusion [81], and experimental acute myocarditis [91]. These results were not reproduced in an in vivo model of acute pancreatitis; in fact, male Wistar rats showed no significant change in serum TNF-α levels following OLE administration [89]. Such a discrepancy cannot be explained with a look at used OLE doses but might be analyzed in the light of the considered outcomes of the studies. Depending on the source of TNF-α, the cytokine concentration might reach a variation detectable only at local level or at both plasma and tissue levels, leading to opposite interpretations about the ability of OLE to reduce TNF-α release. These data do not provide any conclusive detail about local and/or systemic action of OLE.

In vivo data for HT are more consistently reproduced. 200 mg/L HT reduced circulating TNF-α levels in cyclophosphamide-induced immunosuppressed broilers [94] and in a mouse model of LPS-induced systemic inflammation [98], and HT administration diminished serum TNF-α in apoE-/- mice together with a reduction in p38 MAPK and NF-κB phosphorylation in liver [88]. HT-rich extracts reduced TNF-α secretion in both plasma and liver in an in vivo model of high-fat diet [97]. Coherently, in a model of acute liver injury, TNF-α transcription was impaired by HT treatment at tissue level [58].

As recorded in vivo for serum IL-6, also for OLE- and HT-mediated reduction in extracellular release and serum TNF-α, a link with the suppression of MAPK and NF-κB pathways seems to exist [57,66,80,91,96,99].

From a mechanistic point of view, pre-treatment of senescent human neonatal lung fibroblast with 1 μM HT and 10 μM OLE aglycone for 4–6 weeks abolished TNF-α-induced NF-κB nuclear localization and TNF-α triggered signs of inflammation vs. senescent untreated and unstimulated cells [62]. However, it seems that HT and OLE are not able to alter TNF-α-induced NF-κB phosphorylation even at very high concentrations (500 μM), as demonstrated in murine osteoblast-like cells [65].

Use of OLE in humans to ameliorate insulin sensitivity in overweight subjects led to no alterations in serum TNF-α levels [109].

### 2.4. Other Pro-Inflammatory Cytokines

A small number of reports document the effects of OLE and HT on the regulation of other pro-inflammatory cytokines, although the modest amount of evidence hampers the elaboration of conclusive remarks.

IL-2 is fundamental for lymphocyte proliferation and survival and is involved in T cell differentiation and homeostasis [113,116]. Although data about circulating levels of IL-2 in aged subjects are still awaiting a definitive estimation, genetic studies suggest that a rise in IL-2 levels may predispose the subject to unsuccessful aging [102,113]. In vitro data revealed no effect of OLE and HT on IL-2 secretion. On whole blood cell culture obtained from young male volunteers (age range 18–25), 0.1 μM OLE exerted no effect on IL-2 production [74]. Similarly, 50 μg/mL, 25 μg/mL, 12.5 μg/mL, and 6.25 μg/mL HT did not cause any statistically significant difference in IL-2 release by mouse spleen lymphocytes [57]. In vivo evidence for HT is extremely limited. 200 mg/L HT increased IL-2 transcription in both HT-only treated broilers and in cyclophosphamide-induced immunosuppressed broilers vs. cyclophosphamide-induced immunosuppressed broilers that were not receiving HT [94].

IL-8 is a chemotactic factor for neutrophils whose production can be induced by IL-1β and TNF-α [117,118,119] and which has a documented role in chronic inflammation, although its involvement in inflammaging is less defined [113,115]. OLE and HT reduced NF-κB phosphorylation and nuclear localization, thus reducing IL-8 expression and release in vitro [67,73,75,76].

The IL-17 family includes pro-inflammatory cytokines of whom IL-17A is a marker of a specific CD4+ T cell subset named CD4+ T helper 17 (Th17) and is involved in autoimmune diseases as well as infections [16,113,120,121,122]. IL-17A has been reported to be increased in old adults, whereas in centenarians, IL-17A levels were similar to those detected in young adults [114]. In vitro OLE reduced IL-17 expression in ulcerative colitis colonic cells [77]. In an in vivo model of AOM/DSS-induced CRC, only the highest tested levels (100 mg/Kg) of OLE produced a reduction in tissue IL-17A levels [93]. Similarly, in a model of pristane-induced systemic lupus erythematosus, administration of 100 mg/kg HT reduced the amount of IL-17A secreted by LPS-stimulated splenocytes and macrophages [87].

Interferon γ (IFN-γ) is fundamental in orchestrating both innate and adaptive immune responses, with a key role in macrophage activation and in the stabilization of Th1 cells [123,124,125,126]. IFN-γ levels are increased in old adults but even more in centenarians [114,115]. Accordingly, IFN-γ responsive genes were upregulated in centenarians [127], and IFN-γ+ cells among memory and effector CD8+ T lymphocytes increased with age, although no specific genetic explanation was identified for this phenomenon [102]. Thus, the relationship of IFN-γ with longevity is difficult to explain. Low-dose OLE (0.1 μM) had no effect on IFN-γ release in vitro [74]. By contrast, as demonstrated in mouse spleen lymphocytes, treatment with 25 μg/mL, 12.5 μg/mL, and 6.25 μg/mL HT determined an increase in IFN-γ production [57]. Data obtained in vivo are even scarcer. In an in vivo model of colitis-associated colorectal cancer, 50 and 100 mg/kg OLE caused a marked reduction in tissue IFN-γ [93].

### 2.5. Anti-Inflammatory Cytokines

Few reports explore the role of OLE and HT in regulating production and secretion of anti-inflammatory cytokines.

IL-4 is involved in the suppression of inflammation directly interfering with pro-inflammatory cytokine production in macrophages [128,129,130,131] and promoting the differentiation of T helper type 2 (Th2) cells (involved in allergic responses and in the secretion of anti-inflammatory mediators) while suppressing the differentiation of inflammation-promoting Th1 cells [128,129,130,131,132,133]. As a typical Th2-produced cytokine, IL-4 is a master regulator of allergic reactions and asthma but is also involved in immune responses against extracellular parasites [102,128,129,130,131,132,133]. The association of IL-4 with aging has been poorly dissected in literature or produced inconclusive results when assayed experimentally [102,115]. However, IL-4 seems to exhibit a neuroprotective effect during aging [134]. OLE had no effect on IL-4 production when added to whole blood samples taken from young male donors at the concentration of 0.1 μM [74], but in vivo, 20 mg/Kg OLE reduced IL-4 secretion in airway inflammation triggered by cigarette smoke [83]. In mouse spleen lymphocytes, treatment with 50 μg/mL, 25 μg/mL, 12.5 μg/mL, and 6.25 μg/mL HT produced an increase in IL-4 secretion [57]. Consistently, in vivo data showed that 200 mg/L HT upregulated IL-4 at duodenal level in both HT-only treated and in cyclophosphamide-induced immunosuppressed broilers [94] and that 100 mg/kg HT increased IL-4 serum levels in a model of acute liver injury [58].

IL-10 is the most studied anti-inflammatory and immunomodulatory cytokine, exerting a broad immunosuppressant function at multiple levels on both innate and adaptive immunity responses, including Th2-mediated [102,113,135,136,137,138]. During aging, IL-10 levels are found increased especially in successful aging (centenarians), but the connection with age-associated impairment of immune responses needs to be deepened [102,113,114,139,140]. In vitro, OLE increased IL-10 release by human isolated T cells [78]. Similarly, in vitro, 1 μM HT increased IL-10 production in human PBMCs challenged with Parietaria allergens [79] and 41 μM HT increased IL-10 transcription and cytokine secretion in human monocytes after stimulation with LPS [59]. Effects of OLE administration in vivo are not well understood. In vivo, OLE has no effect on IL-10 serum levels in a model of acute pancreatitis [89], but the same dose reduced plasma IL-10 levels in a model of sepsis-induced myocardial injury with a mechanism relying at least in part on the suppression of NF-κB phosphorylation [92]. By contrast, OLE administration elicited an increase in tissue IL-10 mRNA in a model of acetic acid-induced ulcerative colitis [85] and in tissue IL-10 protein in a model of DSS-induced chronic colitis together with a reduction in p38 MAPK phosphorylation [84]. In vivo, HT increased serum IL-10 levels in a model of liver injury [58] and in apoE-/- mice with a reduction in the phosphorylated forms of p38 MAPK and NF-κB [88].

Transforming growth factor beta (TGF-β) is essential for the maintenance of homeostasis during immune responses and in tissue repair after the resolution of inflammation, but due to its involvement in immune differentiation, it is also associated with pathological entities such as fibrosis and age-related diseases, e.g., atherosclerosis, obesity, and frailty [102,113,141,142,143,144,145,146,147]. However, TGF-β levels are reported as decreased in old adults vs. younger subjects, but restored in octogenarians, nonagenarians, and centenarians [102,113,114]. Results in vitro for OLE are inconclusive. OLE increased TGF-β release by human isolated T cells [78], but OLE-rich extract reduced TGF-β expression in LPS stimulated RAW264.7 macrophages [52]. In vivo, HT reduced TGF-β expression in a model of acetic-acid-induced ulcerative colitis [100] and in irradiation-induced pulmonary fibrosis [101].

### 2.6. ROS

Production of radical and non-radical ROS-like superoxide anions ($O_{2}^{•-}$) and hydrogen peroxide (H_2_O_2_), respectively, is a hallmark of aging and a number of chronic and acute diseases, including diabetes mellitus and cancer [32,35,148,149]. The unbalanced increase in ROS may promote structural changes, inflammation, and the establishment of an environment favoring senescence. Such a frame is further complicated by the reduction in master regulators of redox homeostasis and stress response happening during aging, as mentioned below [3,14,31,32,33,34,35,36,148,149].

Both HT and OLE directly scavenge free radicals [46,150,151,152], but part of HT anti-cancer effects is exerted through the induction of apoptosis as a consequence of triggered ROS production [46]. Thus, the ability of HT to promote or counteract ROS production may be strictly dependent on the analyzed context.

OLE and HT showed a concentration-dependent ability to reduce ROS production in human granulocytes challenged with phorbol myristate acetate (PMA) [53]. HT showed an identical ability also on monocytes in the same culture conditions [53].

Experiments on pheochromocytoma PC12 cells [153] and human PBMCs [154] demonstrated that doses of HT ranging from 25 up to 100 μM reduced levels of hypoxia-induced intracellular ROS and protected cells against oxidative damage mediated by 2,3,7,8-Tetrachlorodibenzo-p-dioxin with a mechanism relying on the increase in antioxidant enzyme superoxide dismutase (SOD), catalase (CAT), and glutathione peroxidase (GSH-Px) activity through the PI3K/Akt/mTOR-HIF-1α pathway, with the augmentation of PI3K, phospho-Akt (p-Akt), and phospho-mTOR (p-mTOR) protein levels [153,154]. In addition, HT protects Jurkat cells from H_2_O_2_-induced apoptosis, reduced oxidative stress-induced JNK, and p38 MAPK phosphorylation [155].

In an in vivo model of myocardial ischemia/reperfusion, 20 mg/Kg OLE increased SOD and reduced glutathione (GSH), while diminishing lipid peroxidation marker malondialdehyde (MDA) levels through a mechanism involving suppression of p-IκBα, p53, p-MEK, and p-ERK protein expression [81]. Similarly, during LPS-induced sepsis in mice, pre-treatment with OLE ameliorated the increased levels of MDA and the decrease in cellular antioxidant GSH in liver and kidney [80]. In addition, OLE and HT increase CAT and SOD expression and activity in high-fat diet animal models in liver [97] and adipose tissue [156]. Analogous results about OLE and HT mediated increase in tissue expression/activity of CAT and SOD, and in tissue, GSH and MDA contents were obtained in in vivo models of heart failure [82], cyclophosphamide and epirubicin combined treatment toxicity [90], unilateral ureteral obstruction [95], ulcerative colitis [85], and in cyclophosphamide-induced immunosuppressed broilers [94].

During aging, the main regulator of cellular ROS homeostasis nuclear factor E2-related factor 2 (Nrf2) and the expression of Nrf2 target genes tend to decrease, contributing to the decline in cellular oxidative stress compensation capacities [33,157]. In vitro, HT promoted Nrf2 nuclear localization in RAW264.7 macrophages [53]. In vivo, OLE increased Nrf2 protein expression [158,159] while reducing the expression of Nrf2-regulated gene heme oxygenase-1 (HO-1, an antioxidant defense marker) [96,160,161]. Similarly, in vivo, HT augmented Nrf2 expression and transactivation [32,87,150,157,159,162,163,164]. Nrf2-dependent gene expression mediates oxidative stress response in the presence of HT, including the increase in glutathione-S-transferase (GST), γ-glutamyl cysteine synthetase (γ-GCS), nicotinamide adenine dinucleotide phosphate (NQO1), sirtuin 1 (SIRT-1), and paraoxonase-2 (PON2) mRNAs and HO-1 protein (all of them belonging to cell antioxidant detoxification systems) [32,87,150,157,159,162,163,164].

The use of different means may produce OLE-dependent different effects. In an in vivo model of acute colitis, a reduction in ROS production was obtained only by vehiculating OLE inside cells with nanostructured lipid carrier, whereas OLE in deionized water suspension was more effective in reducing myeloperoxidase activity [68].

## 3. Discussion

Dietary supplementation and intervention may represent a helpful strategy to prevent chronic diseases and ameliorate the associated pathological frame [9,41,42,43,44,45,165]. The contribution of chronic inflammation to unsuccessful aging has been largely documented. However, a comprehensive evaluation of potential applications of olive oil polyphenols in this systemic low-grade inflammatory context has never been performed, with experimental evidence limited to standard scenarios of inflammation [47,48,49]. Interestingly, from data reported in this review, a general poor consistency of OLE-mediated action can be detected, whereas HT results were more frequently reproduced across different in vivo models. Concerns remain about evidence of a HT-mediated in vitro activation of pro-inflammatory molecular circuits whenever HT was administered concomitantly with LPS used as an inflammatory stimulus [57,58,59,60,61,70,71,72]. The biological meaning of HT-dependent increase in IL-1β and TNF-α levels detected in vitro only when HT was not used as a pre-treatment before LPS stimulation remains to be determined. Further investigation would also be necessary to assess why HT-mediated reduction in IL-6 in vitro is not affected by the described phenomenon [58,59]. Nonsteroidal anti-inflammatory drugs interfere with prostaglandin production via the inhibition of the two cyclooxygenase isoenzymes, COX-1 and COX-2 [166], and such a mechanism has historically been correlated with an augmented production of TNF-α [70,167,168]. HT action may exhibit similarities to that exerted by nonsteroidal anti-inflammatory drugs, since HT reduces COX-2 expression in PBMCs sampled from a mouse model of LPS-triggered systemic inflammation [98], LPS-challenged human monocytes [71], THP-1 cells after LPS stimulation [72], LPS-stimulated RAW 264.7 cells [58], and in phorbol myristate acetate (PMA)-stimulated PBMCs and U937 monocytes [169] (with PMA being an activator of protein kinase C, which in turn is shared with the TLR signaling pattern triggered by LPS) [170,171,172,173,174,175,176]. Further, the addition of prostaglandin E2 to cell cultures abolishes HT-dependent effects on TNF-α synthesis and release [70]. Such a frame leaves the space for speculations about the reliability of some experimental systems in order to obtain conclusive evidence about the effects of phenolic compounds in the absence of all biological variables. In addition, doubts persist on the potential equivalent net effect of the olive oil-derived phenols OLE and HT, which are sometimes tested together in the same experimental scenarios [53,62]. Moreover, the potential applicability of HT as a therapeutic rather than a prophylactic agent should be deepened beyond results discouraging its administration out of the limits of pre-treatment before inflammation is established [57,58,59,60,61,70,71,72].

One possible perspective is that during inflammaging, the inflammatory frame starts at local level and gradually becomes systemic [19,177,178,179]. In order to consider the use of OLE and HT as geroprotective agents and/or as therapeutic means once that inflammaging has been established, it would be essential to define if OLE and HT exert their action only at local level. In fact, no conclusive evidence confirms that both phenolic compounds are able to interfere with serum cytokine levels [68,89]. Similarly, the hypothesis that OLE and HT might revert the deregulation of pro-inflammatory cytokine secretion happening as a consequence of age-related impaired ROS homeostasis is still awaiting a resolutive answer.

Aging-associated inflammation should not only be considered as the consequence of environmental/pathological insults. A more complete definition of inflammaging would better be offered in the form of an out-of-context activation of inflammatory mediator secretion ruled by factors such as NF-κB and mTOR, whose action may not automatically overlap that observed in younger subjects [20,21,22,23,24,34,62]. The link between OLE and HT use and activation of pathways accounting for cellular senescence and inflammation [34,35,36] still needs to be elucidated specifically in aged subjects, thus observing putative differences with younger donors.

OLE and HT action were tested in multiple models of LPS-induced sepsis and LPS-challenged cells [52,53,54,57,58,59,63,70,71,80,87], but circulating LPS levels encountered in age-associated gut dysbiosis and alteration of the intestinal barrier are definitely lower than those detected in sepsis or infections (levels that are reproposed in experimental models of LPS-triggered inflammation) [37,38,39]. The effect of OLE and HT on gut-microbiome-derived LPS-driven low-grade inflammation typical of aged subjects is still awaiting a definitive demonstration.

In addition, the role of immune cells in this scenario should not be underrated. Immunosenescence impairs the ability of immune cells to mount effective defenses while removing the homeostatic regulation of immune responses [2,4,5,6,7,8,9,10,11,12]. The potentiality of HT and OLE to revert immunosenescence should be assayed in order to dissect the chance to trigger a rejuvenation of the immune system and to reach an acceptable control of the inflammatory state.

Finally, despite the experimentally documented anti-inflammatory effect of OLE and HT, perplexity persists about possible systemic as well as side effects of both molecules. As regards safety, making an exception for those data documenting OLE- and HT-mediated apoptosis/cell cycle arrest in cancer [46,47,48,52,73,93,164], both molecules had no toxicity and effects on cell cycle on non-cancerous cells/tissues at all [46,64,67,69,71,76,78,79,100], or at the same concentrations used to induce cell death in cancer cells [73]. Only sporadically was a decrease in cell viability reported for non-tumor cells and tissues at extremely high concentrations (≥50 μM for OLE, 200 and 400 μM for HT) [55,60]. In addition, a specific protective effect of OLE and HT against heart, liver, and kidney damage was reported in vivo [46,80,81,82,90,92,95,96,97,151,154,157,164]. However, evidence about OLE and HT effects on human health is mainly limited to olive oil or polyphenol combination ingestion [44,164] and deserves to be deepened especially in terms of potentially harmful consequences, as demonstrated by the unwanted increase in serum IL-6 when OLE was used as a supplement in humans [109].

Absorption of olive oil phenols is estimated to be >60% reaching 95% [180], but in terms of bioavailability and safety of OLE and HT in the elderly, there is a general paucity of data, especially in those cases suffering liver or renal deficiency and/or experiencing polypharmacy [181]. Differences detected when comparing in vivo and in vitro results should be further explored in the light molecular transporters, pharmacodynamics processes, and intracellular targets involved in assimilation, distribution, and metabolism of OLE and HT, and how all these actors change during aging.

## Figures and Tables

**Figure 1 ijms-24-00380-f001:**
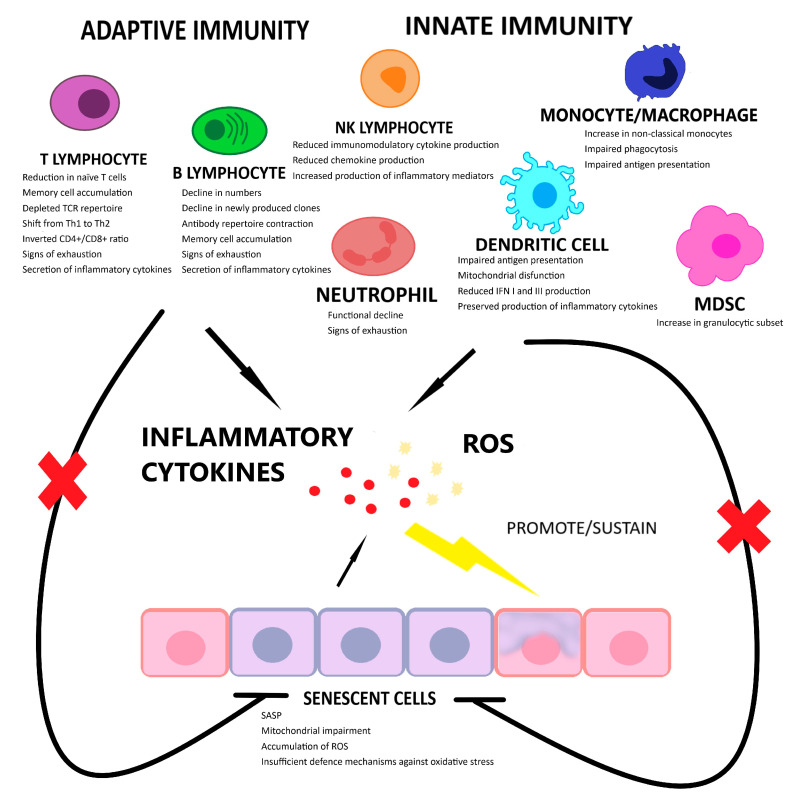
Schematic representation of immunosenescence and senescence triggering and supporting inflammaging. The inability of senescent cells to cope with an age-dependent increased burden of oxidative stress determines the accumulation of ROS and the release of inflammatory mediators that create the conditions for a further expansion of the senescent population. At the same time, immunosenescence determines an increase in both extracellular inflammatory mediators and a reduction in immune cell ability to clear senescent cells. The combination of these events leads to/supports inflammaging. TCR, T cell receptor; IFN, interferon; MDSC, myeloid derived suppressor cell; ROS, reactive oxygen species.This image was realized with Microsoft Paint 3D (Microsoft Corporation, Redmond, WA, USA).

**Table 1 ijms-24-00380-t001:** Effects elicited by OLE and HT on pro- and anti-inflammatory cytokines in vitro.

Cytokine	Molecule	Model	Stimuli	Effect	Ref.
**IL-1β**					
	OLE				
		RAW264.7	LPS	Reduction	[52]
		RAW264.7	LPS	None	[53]
		Human whole blood	LPS	Reduction	[54]
		Human osteoarthritic chondrocytes	None	Reduction	[55]
		RAW264.7	LPS	Reduction	[56]
	HT				
		RAW264.7	LPS	None	[53]
		RAW264.7	LPS	Increase	[57]
		RAW264.7	LPS	Reduction	[58]
		Human monocytes	LPS	Reduction	[59]
		Sprague–Dawley rat chondrocytes	LPS	Reduction	[60]
		Human PBMCs	TNF-α	Reduction	[61]
**IL-6**					
	OLE				
		RAW264.7	LPS	Reduction	[52]
		Human whole blood	LPS	None	[54]
		Human osteoarthritic chondrocytes	None	Reduction	[55]
		RAW264.7	LPS	Reduction	[56]
		Pre-senescent human fetal lung fibroblasts	None	Reduction	[62]
		Pre-senescent human neonatal lung fibroblasts	None	Reduction	[62]
		RAW264.7	LPS	Reduction	[63]
		ARPE-19	IL-1β	Reduction	[64]
		MC3T3-E1	TNF-α	Reduction	[65]
		J774A.1	LPS	Reduction	[66]
		Neonatal human dermal fibroblasts	8 Gy γ-irradiation	Reduction	[67]
	HT				
		RAW264.7	LPS	Reduction	[58]
		Human monocytes	LPS	Reduction	[59]
		Sprague–Dawley rat chondrocytes	LPS	Reduction	[60]
		Pre-senescent human fetal lung fibroblasts	None	Reduction	[62]
		Pre-senescent human neonatal lung fibroblasts	None	Reduction	[62]
		MC3T3-E1	TNF-α	Reduction	[65]
		Neonatal human dermal fibroblasts	8 Gy γ-irradiation	Reduction	[67]
**TNF-α**					
	OLE				
		RAW264.7	LPS	None	[53]
		Human whole blood	LPS	None	[54]
		Human osteoarthritic chondrocytes	None	Reduction	[55]
		RAW264.7	LPS	Reduction	[63]
		J774	LPS	None	[68]
		Human PBMCs	LPS	Reduction	[69]
	HT				
		RAW264.7	LPS	None	[53]
		RAW264.7	LPS	Increase	[57]
		ICR mouse spleen lymphocytes	None	Increase	[57]
		RAW264.7	LPS	Reduction	[58]
		Human monocytes	LPS	Reduction	[59]
		Human monocytes	LPS	Increase	[70]
		Human monocytes	LPS	Increase	[71]
		THP-1	LPS	Reduction	[72]
		HCT116	LPS	Reduction	[73]
		LoVo	LPS	Reduction	[73]
**IL-2**					
	OLE				
		Human whole blood	None	None	[74]
	HT				
		ICR mouse spleen lymphocytes	None	None	[57]
**IL-8**					
	OLE				
		Neonatal human dermal fibroblasts	8 Gy γ-irradiation	Reduction	[67]
	HT				
		Neonatal human dermal fibroblasts	8 Gy γ-irradiation	Reduction	[67]
		HCT116	LPS	Reduction	[73]
		LoVo	LPS	Reduction	[73]
		Caco-2	IL-1β	Reduction	[75]
		Human keratinocytes	IL-1β	Reduction	[76]
**IL-17**					
	OLE	Human ulcerative colitis colonic cells	LPS	Reduction	[77]
**IFN-γ**					
	OLE				
		Human whole blood	None	None	[74]
	HT				
		ICR mouse spleen lymphocytes	None	Increase	[57]
**IL-4**					
	OLE				
		Human whole blood	None	None	[74]
	HT				
		ICR mouse spleen lymphocytes	None	Increase	[57]
**IL-10**					
	OLE				
		Human isolated T cells	None	Increase	[78]
		Human rheumatoid arthritis isolated T cells	None	Increase	[78]
	HT				
		Human monocytes	LPS	Increase	[59]
		Human PBMCs	Parietaria allergens	Increase	[79]
**TGF-β**					
	OLE				
		RAW264.7	LPS	Reduction	[52]
		Human isolated T cells	None	Increase	[78]

IL, interleukin; OLE, oleuropein; HT, hydroxytyrosol; LPS, lipopolysaccharide; PBMCs, peripheral blood mononuclear cells; TNF-α, tumor necrosis factor-α; BPA, bisphenol A; Ref., references.

**Table 2 ijms-24-00380-t002:** Effects elicited by OLE and HT on pro- and anti-inflammatory cytokines in vivo.

Cytokine	Molecule	Model	Experimental Conditions	Effect	Ref.
**IL-1β**					
	OLE				
		BALB/c mice	LPS-induced sepsis	Reduction	[80]
		Sprague–Dawley rats	Myocardial ischemia/reperfusion	Reduction	[81]
		Sprague–Dawley rats	Heart failure	Reduction	[82]
		BALB/c mice	Cigarette-smoke-induced pulmonary inflammation	Reduction	[83]
		C57BL/6 mice	DSS-induced chronic colitis	Reduction	[84]
		Albino rats	Acetic-acid-induced ulcerative colitis	Reduction	[85]
		C57BL/6 J mice	Diet-induced obesity	Reduction	[86]
	HT				
		C57BL/6 mice	LPS-induced acute liver injury	Reduction	[58]
		BALB/c mice	Pristane-induced systemic lupus erythematous	Reduction	[87]
		ApoE-/- mice	Atherosclerosis	Reduction	[88]
**IL-6**					
	OLE				
		BALB/c mice	LPS-induced sepsis	Reduction	[80]
		Sprague–Dawley rats	Myocardial ischemia/reperfusion	Reduction	[81]
		C57BL/6 J mice	Diet-induced obesity	Reduction	[86]
		Wistar rats	Acute pancreatitis	None	[89]
		Sprague–Dawley rats	Epirubicin and cyclophosphamide toxicity	Reduction	[90]
		Lewis rats	Experimental autoimmune myocarditis	Reduction	[91]
		Wistar rats	Sepsis-induced myocardial injury	Reduction	[92]
		C57BL/6 mice	AOM/DSS-induced CRC	Reduction	[93]
		C57BL/6 mice	DSS-induced acute colitis	Reduction	[68]
	HT				
		C57BL/6 mice	LPS-induced acute liver injury	Reduction	[58]
		BALB/c mice	Pristane-induced systemic lupus erythematous	Reduction	[87]
		ApoE-/- mice	Atherosclerosis	Reduction	[88]
		Cobb 500 broilers	Cyclophosphamide-induced immunosuppression	Reduction	[94]
**TNF-α**					
	OLE				
		BALB/c mice	LPS-induced sepsis	Reduction	[80]
		Sprague–Dawley rats	Myocardial ischemia/reperfusion	Reduction	[81]
		Sprague–Dawley rats	Heart failure	Reduction	[82]
		C57BL/6 J mice	Diet-induced obesity	Reduction	[86]
		Wistar rats	Acute pancreatitis	None	[89]
		Sprague–Dawley rats	Epirubicin and cyclophosphamide toxicity	Reduction	[90]
		Lewis rats	Experimental autoimmune myocarditis	Reduction	[91]
		C57BL/6 mice	AOM/DSS-induced CRC	Reduction	[93]
		C57BL/6 mice	DSS-induced acute colitis	Reduction	[68]
		Wistar rats	Unilateral ureteral obstruction	Reduction	[95]
		BALB/cN mice	CP-induced kidney injury	Reduction	[96]
		Swiss rats	High-fat-diet-induced lipid metabolism disturbance	Reduction	[97]
	HT				
		C57BL/6 mice	LPS-induced acute liver injury	Reduction	[58]
		ApoE-/- mice	Atherosclerosis	Reduction	[88]
		Cobb 500 broilers	Cyclophosphamide-induced immunosuppression	Reduction	[94]
		Swiss rats	High-fat-diet-induced lipid metabolism disturbance	Reduction	[97]
		BALB/c mice	LPS-induced systemic inflammation	Reduction	[98]
		Swiss rats	BPA-induced hyperlipidemia and liver injury	Reduction	[99]
**IL-2**					
	HT				
		Cobb 500 broilers	Cyclophosphamide-induced immunosuppression	Increase	[94]
		Cobb 500 broilers	None	Increase	[94]
**IL-17A**					
	OLE				
		C57BL/6 mice	AOM/DSS-induced CRC	Reduction	[93]
	HT				
		BALB/c mice	Pristane-induced systemic lupus erythematous	Reduction	[87]
**IFN-γ**					
	OLE				
		C57BL/6 mice	AOM/DSS-induced CRC	Reduction	[93]
**IL-4**					
	OLE				
		BALB/c mice	Cigarette-smoke-induced pulmonary inflammation	Reduction	[83]
	HT				
		C57BL/6 mice	LPS-induced acute liver injury	Increase	[58]
		Cobb 500 broilers	Cyclophosphamide-induced immunosuppression	Increase	[94]
		Cobb 500 broilers	None	Increase	[94]
**IL-10**					
	OLE				
		Wistar rats	Acute pancreatitis	None	[89]
		Wistar rats	Sepsis-induced myocardial injury	Reduction	[92]
		C57BL/6 mice	DSS-induced chronic colitis	Increase	[84]
		Albino rats	Acetic-acid-induced ulcerative colitis	Increase	[85]
	HT				
		C57BL/6 mice	LPS-induced acute liver injury	Increase	[58]
		ApoE-/- mice	Atherosclerosis	Increase	[88]
**TGF-β**					
	HT				
		Albino rats	Acetic acid-induced ulcerative colitis	Reduction	[100]
		Sprague–Dawley rats	Irradiation-induced pulmonary fibrosis	Reduction	[101]

IL, interleukin; OLE, oleuropein; HT, hydroxytyrosol; LPS, lipopolysaccharide; CP, cisplatin; DSS, dextran sulfate sodium; CRC, colorectal cancer; AOM, azoxymethane; BPA, bisphenol A; Ref., references.

## Data Availability

Not applicable.

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
