# Peer review of "Effects of Oleuropein and Hydroxytyrosol on Inflammatory Mediators: Consequences on Inflammaging"

_ijms, 2022, doi:10.3390/ijms24010380_

Round 1
Reviewer 1 Report
The review titled “Effects of Oleuropein and Hydroxytyrosol on Inflammatory Mediators: Consequences on Inflammaging” summarize the recent literatures on nutraceutical interventions using Olive phenolic compounds to ameliorate the systemic inflammation and oxidative stress in ageing. This review is well written but needs some minor changes in the text. Authors should improve the figure and text size in the figure. Additional figures or a table are recommended.
Suggested corrections
1. Aging spelling: aging or ageing
2. Abstract: pro-inflammatory to proinflammatory
3. Page 2, line 7 as for example change to e.g.
4. Figure 1: Better resolution image needed; text size is too small not readable. Please mention if any specific software used for figures e.g., BioRender.com.
5. Page 3 LPS to Lipopolysaccharide (LPS).
6. Page 3 para 5 Olea europaea to Olive (Olea europaea L.).
7. Section 2 please mention the animal models used in the in vivo part.
8. Page 4, paragraph 4: functioning to functions.
9. Section 2.2 please group the effect of OLE and HT separately as in 2.1.
10. In section 2.3 line 8 please describe how TNF regulates proliferation and apoptosis as they are two opposite events.
11. Page 7-line 8 species spelling check.
12. If possible, please include a separate paragraph about the effect of OLE and HT on human.
13. More description on ROS and its role in ageing recommended.
Reviewer 2 Report
The manuscript is well done, but some major points must be rewired.
Item 1- Figure 1- Low resolution. I cannot read the text in the Figure 1. Too small.
Item 2- These products have any hepatotoxic effect. Please show that Oleuropein and Hydroxytyrosol have or not toxic effect on cell line, liver, kidney or other organ.
Item 3- Please insert tables to in vivo and other to in vitro analysis. Showing the following: Study (model or cell type), stimulus (on cell or in animal), drug induced treatments Results and references
Item 4- Please change the reference 1.
Round 2
Reviewer 2 Report
The manuscript is corrected and now it can be published in this journal